# Profiling and Evaluation of the Effect of Guarana-Loaded Liposomes on Different Skin Cell Lines: An In Vitro Study

Isabel Roggia [1,†], Patrícia Gomes [1,†], Ana Julia Figueiró Dalcin [1], Aline Ferreira Ourique [1], Ivana Beatrice Mânica da Cruz [2], Euler E. Ribeiro [3], Montserrat Mitjans [4] and Maria Pilar Vinardell [4,*]

[1] Laboratory of Nanotechnology, Franciscan University, Santa Maria 97010-032, Rio Grande do Sul, Brazil; isa_roggia@yahoo.com.br (I.R.); patriciagomes@ufn.edu.br (P.G.); anajuliadalcin@hotmail.com (A.J.F.D.); alineourique@gmail.com (A.F.O.)

[2] Laboratory of Biogenomic, Federal University of Santa Maria, Santa Maria 97105-900, Rio Grande do Sul, Brazil; ibmcruz@hotmail.com

[3] Center for Research, Teaching and Technological Development (Gerontec/FUnATI), Manaus 69029-040, Amazonas, Brazil; unatieuler@gmail.com

[4] Departament de Bioquimica i Fisiologia, Facultat de Farmàcies i Ciencies de l'Alimentació, Universitat de Barcelona, Av. Joan XXIII 27-31, 08028 Barcelona, Spain; montsemitjans@ub.edu

* Correspondence: mpvinardellmh@ub.edu

† These authors contributed equally to this work.

**Abstract:** The objective of this study was to analyze the in vitro stability and toxicity of liposomes containing guarana in skin cell lines. The liposomes were produced by the reverse phase evaporation method containing 1 mg/mL guarana. The stability of the liposomes was evaluated by physical-chemical parameters for up to 90 days using three different storage conditions. The cytotoxicity of guarana (GL), liposomes (B-Lip), and guarana-loaded liposomes (G-Lip) was evaluated on spontaneously immortalized human keratinocyte cell lines (HaCaT), murine Swiss albino fibroblasts (3T3), and human fibroblasts (1BR.3.G). The evaluation was performed using cellular viability analysis. The techniques used were 3-(4,5-dimethylthiazol-2-yl)-2,5-diphenyltetrazolium bromide (MTT) and neutral red capturing (NRU), and the analyses were conducted after 24, 48, and 72 h of exposure of these cells to the different treatments. The G-Lip exhibited physical-chemical stability for 60 days when the samples were stored in a refrigerator. The GL, B-Lip, and G-Lip demonstrated low cytotoxicity in the three different cell cultures tested since a small reduction in cell viability was only observed at the highest concentrations. In addition, greater cell damage was observed for B-Lip; however, guarana protected the cells from this damage. Thus, G-Lip structures can be considered promising systems for topical applications.

**Keywords:** *Paullinia cupana*; natural products; nanoparticles; stability; cytotoxicity; cell culture





## 1. Introduction

Plants are important sources of bioactive compounds in modern medicine, and approximately one-third of the best-selling pharmaceuticals are from natural products or their derivatives [1–3]. In contrast, many natural active ingredients are unstable compounds that may undergo degradation, oxidation reactions, or both. In addition, these reactions may lead to a decrease in or loss of efficacy of the active compounds. For example, incorrect storage may promote the loss of active compounds, whether for physical or biological reasons [4,5].

One alternative that attempts to resolve these limitations is the incorporation of nanoparticles into natural product-based delivery systems, which increases the stability of the compounds and consequently preserves the therapeutic effects using techniques that involve nanotechnology [6–8]. These nanoparticles can significantly increase both the in vitro and in vivo bioavailability of natural products [1]. From this perspective, the

pharmaceutical industry has increasingly used nanotechnology-based products to create cosmetic formulations [9–12].

Liposomes are among the nanoparticles used for the development of nanocosmetic products [4,13–15]. Liposomes are structures formed by a lipid bilayer that can fuse with the layers when applied to the skin, thereby promoting the release of active compounds, making them useful as carriers for cosmetic applications [16]. In addition to having a simple means of preparation, the use of these systems can promote greater absorption of the active compounds into the skin, so they constitute an interesting system when discussing cosmeceuticals. The enhanced absorption is associated with prolonged release, thereby promoting a greater effect [9,17,18]. The lipid structure of the liposomes facilitates the fusion of active compounds with different layers of the skin. This process is more advantageous than other nanostructured systems in the transport of nanocosmetics, such as vitamins, minerals, antioxidants, and anti-aging materials, making it a useful tool in the field of pharmaceuticals and cosmetics [9,19,20].

Studies of the interactions of these nanoparticles with biological systems, such as their bioavailability, biodegradability, and toxicity, are of the utmost importance. Hence, it is essential to know the physicochemical properties of these particles, such as their size, shape, surface area, morphology, and stability [21]. In addition, precise and predictive risk assessment approaches are required for understanding the potential health and environmental hazards associated with exposure to nanomaterials [22].

The use of animals in scientific research is the most commonly used method to ensure safety and low-level toxicity. However, the increase in ethical discussions and regulatory standards regarding the protection of animals used for scientific purposes (Directive 2010/63/EU) [23,24], as well as the growing interest in the search for predictive toxicology, has been changing the perspective in this line of research [25–28]. Russell and Burch (1959) [29] postulated the 3Rs principle, which is primarily aimed at the reduction, refinement, and replacement of laboratory animals. Researchers have stated that good science and animal welfare must go hand in hand. Hence, several alternative methods have been proposed in an attempt to reduce the number of animals used in experimentation and the cost of experiments [24,30,31].

In this context, several skin models have been designed to predict the interactions of chemical compounds and/or nanoparticles with skin barriers by mimicking the skin's physiological barriers. Among these models, cellular models are the oldest and best described in the literature. For skin-related toxicity studies, the most commonly used cell types are immortalized human keratinocyte cells (HaCaT) and fibroblasts (3T3) because keratinocytes are the main cells of the epidermis, and fibroblasts represent the main type of cell in the dermis and are involved in the production of constituents of the extracellular matrix, such as collagen, glycosaminoglycans, and proteoglycans. In addition, HaCaT cells, for example, are low-cost cells, with great ease of use and rapid cell proliferation, in addition to high experimental reproducibility, and they express the main skin surface markers and the functional activity of isolated keratinocytes [32,33]. Fibroblasts (human or murine) represent the most used cell type in cell culture because they are easily cultivated and maintained in vitro. 3T3 fibroblasts are cells from Swiss mouse embryos in which the fibroblasts have been transformed with the SV40 virus T antigen into a stable growing cell line. Similarly, human fibroblasts can be easily isolated from different body sites and easily cultured in vitro [34]. Therefore, in our study, keratinocytes (HaCaT) and fibroblastic cells (3T3 and 1BR.3.G) were chosen as the model system for the epidermis and dermis, respectively.

Guarana, *Paullinia cupana var. sorbilis (Mart.) Ducke (Sapindaceae)*, is a native Brazilian species of considerable economic and social importance [35]. Among the Amazonian species, guarana is one of the most promising species in the Brazilian flora [36]. Guarana has a long history of use as a stimulant, mainly by indigenous tribes in Brazil, and it is a versatile plant due to its potential utility in the food industry, such as in the preparation of energy drinks, soft drinks, and food supplements [37–40]. Furthermore, guarana is

widely used in the pharmaceutical industry in the production of drugs and is listed in the Brazilian Pharmacopoeia [41]. It is present in several cosmetic products due to its antimicrobial [42,43] and antioxidant activities [37–39,42,44–53]. Due to the high content of alkaloids in guarana, extracts are added to products for the treatment of gynoid lipodystrophy and to anti-aging creams [54,55]. Given the importance of guarana and the increasing use of its seeds, there has been increased interest in the quality of the products containing this compound because its chemical structure is predominantly unsaturated and susceptible to oxidation [56].

In the present study, we evaluated the physicochemical stability of liposomes containing 1 mg/mL guarana powder by reverse phase evaporation. Moreover, based on the potential topical application of these new nanocarriers, the in vitro cytotoxicity of guarana (GL), the blank liposome (without guarana, B-Lip), and the liposome containing 1 mg/mL guarana powder (G-Lip) was tested. The evaluations were conducted in different cultures of skin cells, fibroblasts (3T3 and 1BR.3.G), and keratinocytes (HaCaT).

## 2. Materials and Methods

### 2.1. Materials

Acetonitrile of an analytical standard, dimethyl sulfoxide (DMSO), 3-(4,5-dimethylthiazol-2-yl)-2,5-diphenyltetrazolium bromide (MTT), neutral red dye (NR), cholesterol, and polysorbate 80 were purchased from Sigma-Aldrich® (St. Louis, MO, USA). Methanol and trifluoroacetic acid (TFA) were acquired from J.T. Baker® (Mexico City, Mexico). Ethanol was acquired from Synth® (São Paulo, Brazil). Monobasic potassium phosphate was acquired from F. Maia® (São Paulo, Brazil). Sodium chloride and dibasic sodium phosphate were obtained from Nuclear® (São Paulo, Brazil). Potassium chloride was obtained from Qhemis® (São Paulo, Brazil). Lipoid S100® was obtained from Lipoid® (Ludwigshafen, Germany) and vitamin E from Alpha química® (Porto Alegre, Brazil). Dulbecco's modified Eagle's medium (DMEM), fetal bovine serum (FBS), phosphate-buffered saline (PBS), L-glutamine solution (200 mM), trypsin-EDTA solution (170,000 U/L trypsin and 0.2 g/L EDTA), and penicillin-streptomycin solution (10,000 U/mL penicillin and 10 mg/mL streptomycin) were obtained from Lonza (Verviers, Belgium). The 75-cm$^2$ flasks and 96-well plates were obtained from TPP® (Trasadingen, Switzerland). Guarana powder was kindly provided by Agropecuary Research Brazilian Enterprise (EMBRAPA Western Amazon in Manaus, Amazon, Brazil).

### 2.2. Preparation and Characterization of Guarana-Loaded Liposomes

Liposomes containing 1 mg/mL guarana powder (G-Lip) were prepared by reverse phase evaporation [57,58] after being previously developed and standardized by our research group [59]. The soy phosphatidylcholine (0.8 g), cholesterol (0.15 g), and vitamin E (0.02 g) were solubilized in ethanol (40 mL) with the aid of ultrasound for 5 min. Then, an aliquot of an aqueous solution (4 mL) of guarana powder (0.1 g) and polysorbate 80 (0.15 g) in PBS pH 7.4 was sonicated for 5 min, thereby yielding a dispersion of reverse micelles. The organic solvent was removed by evaporation to form an organogel. The organogel reverted to vesicles after the addition of the remainder of the aqueous phase by stirring (300 rpm) for 30 min using a rotary evaporator in a water bath at 40 °C. The vesicles were homogenized at room temperature using filtering sequences through the use of 0.45- and 0.22-µm filter membranes (Millex Syringe Filter®). Blank liposomes (B-Lip) were also produced by the same method under the identical experimental conditions previously described, except at this stage, the guarana was suppressed from the formulation.

For the characterization, the average diameter parameters were evaluated by two different techniques: laser diffraction (Microtrac S3500®, EUA) using the undiluted dispersions; and dynamic light scattering (Zetaziser Nano-ZS®, Malvern, United Kingdom) using samples diluted in ultrapure water (1:500 *v/v*). The latter method also determined the polydispersity index (PDI). To determine the homogeneity of the suspended vesicles,

the span index was calculated from the data obtained by laser diffraction analysis, using the following formula:

$$\text{span index} = \left[ \frac{dv\ (90\%) - dv\ (10\%)}{dv\ (50\%)} \right]$$

where: *dv* is the size (μm) 10, 50 and 90%.

The zeta potential values of the liposomes were evaluated by the determination of electrophoretic mobility (Zetaziser Nano-ZS®, Malvern, United Kingdom). The measurements were performed after diluting the formulations in 10 mM NaCl aqueous solution. The pH values of the formulations were directly determined using a calibrated potentiometer (Digimed DM22®, São Paulo, Brazil). The organoleptic characteristics (appearance, color, and odor) were visually evaluated, and the changes in the initial sample (time zero) were observed.

The five main active compounds present in guarana powder (theobromine, theophylline, caffeine, catechin, and epicatechin) were quantified. The quantification end encapsulation/incorporation efficiency was determined by reverse-phase high-performance liquid chromatography (RP-HPLC) using a Prominence® chromatograph (Shimadzu®, Kyoto, Japan), according to methodologies described and validated by our group [49,59].

The chromatographic instruments and conditions were a Shimadzu HPLC system (Kyoto, Japan) equipped with an LC-20AT pump, an SPD-M20A photodiode array (PDA) detector, a CBM-20A system controller, a C18 Phenomenex (4 × 3.0 mm, 5 μm) precolumn, and an RP-18 Phenomenex column (250 mm × 4.5 mm, 5 μm). Water was used as the mobile phase with 0.1% TFA (pH 4.2, A) and a methanol–acetonitrile solution (25:75 *v/v*, B) in a 90:10 *v/v* ratio (A:B) at an isocrative flow rate (1 mL/min). The injection volume was 20 μL. The detection was performed at 280 nm. The tests were based on the methodology described by Klein et al. (2012) [60], and some modifications were validated by our research group [59].

### 2.3. Physicochemical Stability Study of Guarana-Loaded Liposomes

The liposome samples containing 1 mg/mL guarana powder were prepared in triplicates and stored at room temperature (RT 25 ± 2 °C), in a climatic chamber (CC 40 ± 2 °C and 75% relative humidity) and under refrigeration (RE 5 ± 2 °C), and they were analyzed at 0, 7, 15, 30, 60, and 90 days. The parameters analyzed were the organoleptic characteristics (appearance, color, and odor), precipitate formation or phase separation, mean vesicle diameter, polydispersity index, zeta potential, pH, concentration of active compounds, and encapsulation/incorporation efficiency, using the methodology described previously (Section 2.2).

### 2.4. Culture of 3T3, HaCaT and 1BR.3.G Cell Lines

The HaCaT, 3T3, and 1BR.3.G were grown in DMEM (4.5 g/L glucose), supplemented with 10% FBS, L-glutamine (2 mM), penicillin (100 U/mL), and streptomycin (100 μg/mL) at 37 °C in 5% $CO_2$. The cells were routinely cultured in 75-$cm^2$ culture flasks and trypsinized using trypsin-EDTA when the cells reached approximately 80% confluence. The HaCaT cell lines were obtained from the Eucellbank (University of Barcelona, Spain), whereas 3T3 was obtained from ECACC (Sigma-Aldrich®), and 1BR3.G was donated by Prof. Ramon Mangues (Biomedical Research Institute Sant Pau of the Hospital de Sant Pau, Barcelona, Spain).

### 2.5. Analysis of Liposome Interference with Cell Viability Assays

To eliminate the potential interference of the liposomes with the cell viability assays, an interference test was performed prior to the experiments, using the methodology described by Nogueira et al. (2013) [61]. In this analysis, the G-Lip (500 μL) was suspended in DMEM (500 μL, without FBS and phenol red) containing the MTT dye (0.5 mg/mL) or

NR (0.05 mg/mL). These solutions were prepared in triplicate and incubated at 37 °C in 5% $CO_2$.

After 3 h of incubation, the liposomes were centrifuged (10 min at 19,000 rpm). The supernatant was removed, and the DMSO (1 mL) or a solution (1 mL) containing 50% absolute ethanol and 1% acetic acid in distilled water was added for the MTT and NR experiments, respectively. These solutions were shaken and transferred to a quartz cuvette, in which a scanning spectrum was plotted in the range of 300 to 700 nm. The absorbance was recorded at 550 nm using a Shimadzu double beam UV-160A-Vis spectrophotometer (Shimadzu®, Kyoto, Japan).

### 2.6. Cytotoxicity Assays

The cytotoxic effects of GL, G-Lip, and B-lip were measured by MTT tetrazolium salt assay, as described by Mosmann (1983) [62], and neutral red uptake (NRU) assay, as described by Borenfreund and Puerner (1985) [63]. The 3T3, HaCaT, and human fibroblast cells were seeded in the 60 central wells of a 96-well plate at densities of $1 \times 10^5$, $6.5 \times 10^4$, and $5.5 \times 10^4$ cells/mL for 24, 48, and 72 h, respectively. After incubation (24 h, 5% $CO_2$, 37 °C), the medium was removed, and 100 μL of the DMEM supplemented with 5% FBS containing the different treatments at the required concentration (3.91–500 μg/mL) was added. After incubation under identical conditions as before, the medium was removed, and 100 μL of MTT in PBS (5 mg/mL) diluted (in a 1:10 ratio) in DMEM was added without FBS. The phenol red was then added to the cells for a final concentration of 0.5 mg/mL. Similarly, 100 μL of 0.05 mg/mL NR solution in DMEM without FBS and phenol red was added to each well for the NRU assay. The controls used in the experiments consisted of cells and medium without any treatment. The plates were again incubated for 3 h after the medium was removed. Then, for the MTT assay, 100 μL of DMSO was added to each well to dissolve the purple formazan product. For the NRU assay, 100 μL of a solution containing 50% absolute ethanol and 1% acetic acid in distilled water was added. After 10 min on a microtiter plate shaker at room temperature, the absorbance of the resulting solutions was measured at 550 nm using a microplate reader. The cell viability was calculated by considering the mean absorbance of each concentration with respect to that of the controls. The analysis was always performed in triplicate: three wells for each treatment at each concentration, and all experiments were also repeated three times.

### 2.7. Statistical Analysis

All the experiments were evaluated in triplicates. The results are expressed as mean ± standard deviation (SD), and the statistical analyses were performed using one-way analysis of variance (ANOVA) to determine the differences between the datasets, followed by Dunnett's test or Tukey's post hoc test for multiple comparisons using the GraphPad Prism software, version 5.0 ®. The differences were considered significant at $p < 0.05$.

## 3. Results and Discussion

### 3.1. Stability Study of Guarana-Loaded Liposomes

Considering the hydrophilic and lipophilic characteristics of the active compounds present in guarana, liposomes were the nanostructures selected in our study as vehicles for the incorporation of guarana powder because the structures allow for the incorporation of compounds with different characteristics [20,64–67].

In a previous study conducted by our research group [49], we evaluated two different methods of producing liposomes (ethanol injection and reverse phase evaporation) and different concentrations of guarana powder associated with these structures (1, 5, and 10 mg/mL). From this initial study, the reverse phase evaporation method and the concentration of 1 mg/mL were selected as the optimum conditions. Hence, the results herein refer to liposomes produced under the aforementioned conditions (Figure 1).

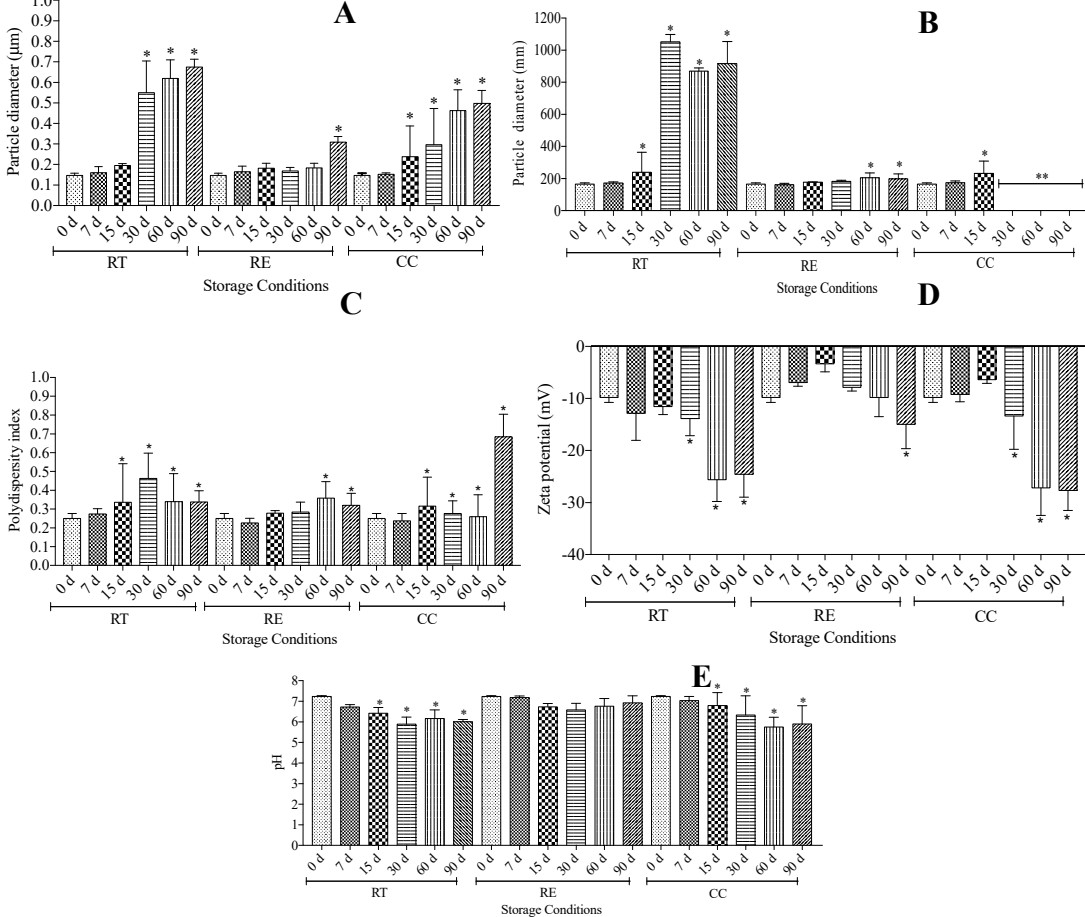

**Figure 1.** (**A**) Distribution of the mean diameter by the laser diffraction technique; (**B**) distribution of the mean diameter by dynamic light scattering; (**C**) polydispersity index; (**D**) zeta potential; (**E**) pH values. Results are expressed as mean ± standard deviation (n = 3). RT (room temperature, 25 ± 2 °C), CC (climatic chamber, 40 ± 2 °C and 75% relative humidity), and RE (under refrigeration, 5 ± 2 °C). * Significant difference ($p < 0.05$) in relation to the initial values. ** micrometric vesicle diameter (μm).

To verify the formation of a homogeneous and nanometric system, the vesicle diameter distribution analysis was performed by the laser diffraction technique (Microtrac®). The initial mean diameter was 147 ± 0.01 nm; this parameter was measured for 90 days under different storage conditions. The diameter remained stable, with no significant differences for 7, 15, and 60 days when stored at CC, RT, and RE, respectively (Figure 1A).

From these results, the *span* index was calculated, with which it was possible to determine the homogeneity of the suspended vesicles. The initial *span* values of 0.31 ± 0.10 indicate close distribution of the vesicles. These values remained low (0.72 ± 0.11, 0.79 ± 0.41, and 0.69 ± 0.31) for up to 90 days when the samples were stored at RT, RE, and CC, respectively. The presence of low dispersion nanometric vesicles was also observed by the dynamic light scattering technique (Figure 1B).

We observed initial vesicles 165 ± 8.27 nm in size and a PDI of 0.250 ± 0.03, which remained largely similar for 7 days when stored in RT (172 ± 3.98 nm, PDI 0.273 ± 0.03) and CC (174 ± 3.03 nm, PDI 0.237 ± 0.02); under this same condition, the destabilization of the system was observed in 30 days of storage in the presence of micrometric vesicles (1.355 μm) (Figure 1B). This increase was associated with alterations in the organoleptic characteristics of appearance, color, and odor at 15 days and was intensified at 30 days (Figure 2).

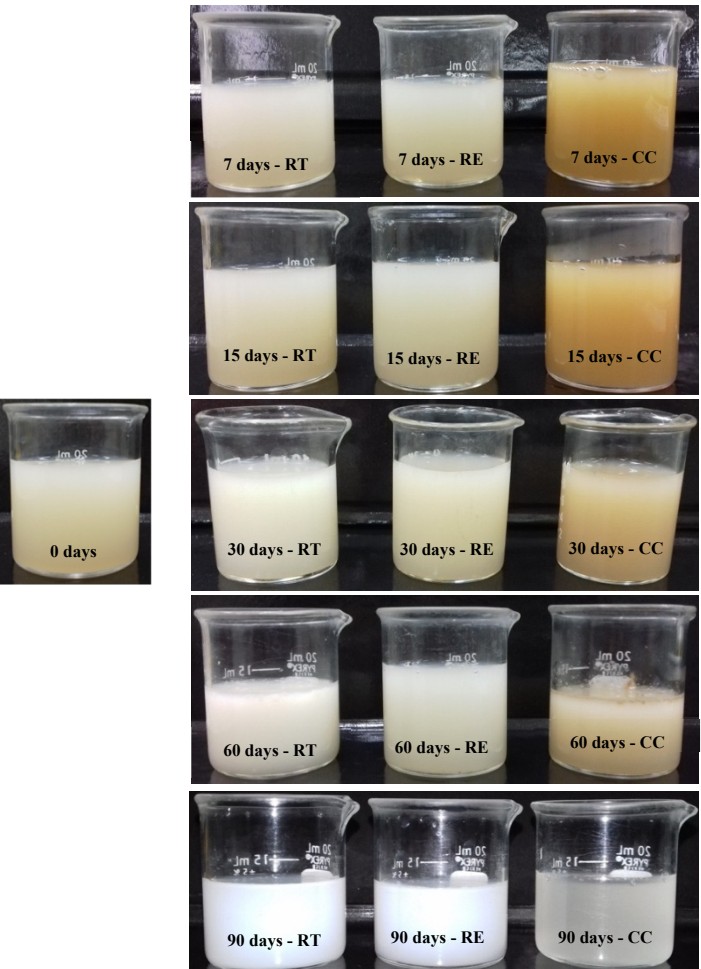

**Figure 2.** Macroscopic characteristics of the liposomes prepared by the reverse phase evaporation method, stored under different storage conditions: RT (room temperature, 25 ± 2 °C), CC (climatic chamber, 40 ± 2 °C and 75% relative humidity), and RE (under refrigeration, 5 ± 2 °C).

In contrast, the vesicles were not significantly different for 30 days when stored under RE (181 ± 4.70 nm, PDI 0.285 ± 0.04) (Figure 1B), whereas changes in the organoleptic characteristics were confirmed at 90 days of analysis (Figure 2).

The formulations were also analyzed for the zeta potential by the electrophoretic mobility technique, using the data presented in Figure 1D. The liposomes had an initial zeta potential of −9.78 ± 0.98 mV and remained without significant differences for 15 days when stored in RT (−11.56 ± 3.47 mV) and in CC (−6.40 ± 1.40 mV). For the samples stored under RE, the initial characteristics were maintained at up to 60 days of storage (−9.81 ± 2.28 mV).

The low zeta potential values are in agreement with the characteristics of the phospholipid (soy phosphatidylcholine) used for the production of these liposomes [68]. The findings in this paper corroborated the results obtained by Karn, Parkl and Hwangl (2013) [69], who produced liposomes using Lipoid S100® and cholesterol and obtained vesicles with potentials of −6.8 to −7.7 mV.

The initial pH values of 7.24 ± 0.04 were maintained without significant differences for up to 7 days of analysis when the samples were stored at RT (6.92 ± 0.04) and in CC (7.03 ± 0.01). For the samples stored under RE, no significant differences were observed over the 90 days (Figure 1E).

The pH values were as expected, based on the preparation of these nanostructures, in which PBS (pH 7.4) was used as the aqueous phase. The pH change associated with higher temperatures (RT and CC) is directly related to the stability of the liposomes. It is believed

that the reduction in pH when the samples were stored under these conditions could be related to the hydrolysis of the lipids present in the liposomal structures, which at high temperatures can undergo chemical degradation, leading to a loss of stability [70,71].

It is known that the evaluation of the organoleptic characteristics is of great importance because physical processes, such as aggregation, flocculation, fusion, or coalescence, can alter the utility of the liposomes, which can result in the loss of the associated liposomes and changes in the sizes of the structures [72,73]. The results regarding the organoleptic characteristics are shown in Figure 2.

In summary, it was observed that the samples stored in CC revealed alterations in color from 7 days of storage (Figure 2). At 15 days, the samples stored at RT and CC exhibited an intense rancid odor, with significant changes in the vesicle diameter and polydispersity index (Figure 1B,C) and pH (Figure 1E). These changes intensified at 30 days of storage, mainly under the CC condition, with changes in the zeta potential as well (Figure 1D). Under these two conditions, complete phase separation was observed after 60 days. For the samples only stored in RE at 90 days, changes in organoleptic characteristics were observed.

Each component, active or not, can affect the stability of a formulation. These alterations can be classified as intrinsic, when they are related to factors inherent to the formulation, such as physical, chemical, pH, reactions, and hydrolysis incompatibility, among others; or extrinsic, when related to factors external to the formulation, such as time, temperature, light, oxygen, humidity, packaging material, microorganism, bacteria, or sample vibration [74]. Among these possibilities, it is believed that the color change at 7 days for the sample stored in CC may be related to processes extrinsic to the formulation; that is, the high temperature at which the sample was conditioned may have triggered and/or accelerated physical-chemical and chemical reactions, resulting in changes in organoleptic characteristics, such as appearance, color, and odor.

It should also be noted that blank liposomes (in the absence of guarana) were produced and characterized at the same times under identical conditions (data not shown). The results were similar to those of the liposomes containing 1 mg/mL of guarana powder; that is, guarana did not alter the characteristics of the liposomal structures.

The total content and encapsulation/incorporation efficiency of the five main active compounds (theobromine, theophylline, caffeine, catechin, and epicatechin) were also evaluated. The results obtained for the total content of the assets are depicted in Figure 3.

The quantification results (Figure 3) indicated the presence of 20.61 μg/mL methylxanthines (0.14 μg/mL (0.028%) TEOB, 0.47 μg/mL (0.094%) TEOF, and 20.00 μg/mL (4.0%) CAF) and 26.00 μg/mL polyphenols (13.00 μg/mL (2.6%) CAT and 13.00 μg/mL (2.6%) EPICAT) in the guarana powder sample.

For TEOB, the initial content of 104.73 ± 1.11% decreased to 64.80 ± 18.11% (15 days) and 63.99 ± 2.06% (90 days) when the liposomes were stored in RT and CC, respectively. When the samples were stored in RE, there were no significant differences in the concentrations until 90 days, with a final concentration of 95.34 ± 1.02% (Figure 3A).

For TEOF, the initial content (91.99 ± 1.07%) significantly changed at 15 days at RT (84.08 ± 58.71%) and in CC (84.24 ± 58.98%). When the liposomes were stored under RE, the decrease in the content of this active compound was only observed at 90 days, with a final content of 40.07 ± 4.93% (Figure 3B).

The CAF demonstrated a reduction in the initial content (100.96 ± 0.59%) after 15 days when the samples were stored at RT (35.55 ± 31.77%) and in CC (62.90 ± 25.71%). The initial content of the CAF exhibited no significant difference at 90 days, when the sample was stored under RE (98.89 ± 4.37%) (Figure 3C).

The polyphenols (CAT and EPICAT) (Figure 3D,E) were the active compounds that demonstrated the greatest reduction in content, which was already observable at 7 days of stability, when the samples were stored in CC. The CAT initially showed a content of 92.90 ± 1.07% and was significantly reduced to 10.65 ± 6.90% under this storage condition. The same outcome was observed for EPICAT, which had an initial content of 85.35% ± 2.99 and, at 7 days, a stability content of 21.51% ± 18.66. When these compounds (CAT and

EPICAT) were at RT, the content was significantly altered at 15 days, with concentrations of $80.70 \pm 0.45$ and $77.90 \pm 3.18\%$ for CAT and EPICAT, respectively. Under refrigeration, the concentration was reduced at 60 days for both active compounds, with content of $63.87 \pm 30.03$ and $53.78 \pm 22.06\%$ for CAT and EPICAT, respectively.

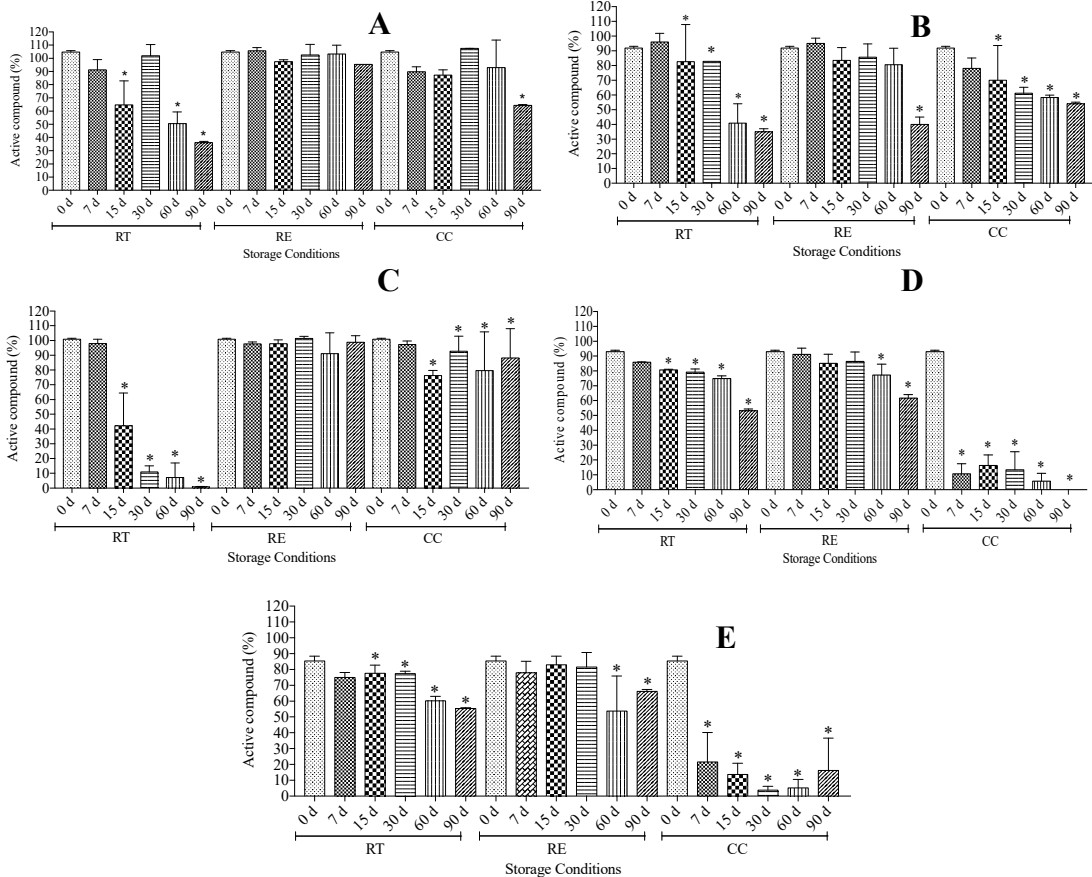

**Figure 3.** Total content of liposomal active ingredients containing 1 mg/mL of guarana powder. Results are expressed as mean $\pm$ standard deviation (n = 3). (**A**) (theobromine), (**B**) (theophylline), (**C**) (caffeine), (**D**) (catechin), and (**E**) (epicatechin). RT (room temperature, $25 \pm 2$ °C), CC (climatic chamber, $40 \pm 2$ °C and 75% relative humidity), and RE (under refrigeration, $5 \pm 2$ °C). * Significant difference ($p < 0.05$) compared to the initial values.

In general, the condition that yielded the highest stability of the active compounds was under RE. For TEOB and CAF, the content remained unchanged throughout the stability study. When stored in RE, the CAT and EPICAT exhibited significant reductions at 60 days of stability, whereas TEOF demonstrated a reduction at 90 days.

The reduction in the content, particularly for the polyphenols (CAT and EPICAT) when the samples were stored in CC, could be caused by the oxidation of these compounds at higher temperatures. In previous studies, the polyphenols present in cocoa exhibited enzymatic oxidation when high temperature and high humidity were used to dry this product [75,76].

The literature also describes the polyphenols as unstable structures that may undergo possible oxidative processes when under neutral and alkaline conditions. This instability was visualized through three degradation processes: decomposition into smaller molecules, polymerization in other molecules, and oxidation to oxidative molecules under natural conditions [77].

In our study, we demonstrated protection against this alkaline degradation when the polyphenols were incorporated into liposomal structures. When the formulations were stored in RE, the content of polyphenols (CAT and EPICAT) was maintained without signif-

icant differences for up to 60 days of stability. In previous studies, when the degradation of these CAT and EPICAT compounds present in guarana was evaluated at baseline alkaline conditions (0.1 M NaOH), the final CAT and EPICAT contents were approximately 62.13% and 23.25%, respectively, when the guarana powder was exposed to this condition for 15 min.

Data from the literature indicated that green tea liposomes containing polyphenols, such as catechin, prepared with lecithin, cholesterol, and phosphate buffer at pH 6.62, exhibited greater stability of this active compound against oxidative processes when compared to non-nanostructured green tea [78].

After the total quantification of the active compounds present in guarana, the encapsulation/incorporation of these active substances was determined in the liposome sample. The results of this evaluation are presented in Table 1.

**Table 1.** Efficiency of encapsulation/incorporation of liposomes containing 1 mg/mL of guarana powder, prepared by the reverse phase evaporation method (n = 2).

| | Reverse Phase Evaporation | |
| Active | Initial (%) * | 90 Days (%) ** |
| --- | --- | --- |
| TEOB | Not determined | Not determined |
| TEOF | Not determined | Not determined |
| CAF | 17.02 ± 0.60 | 30.13 ± 0.23 |
| CAT | 74.34 ± 1.93 | 51.65 ± 0.77 |
| EPICAT | 87.53 ± 0.94 | 70.88 ± 2.17 |

TEOB (theobromine), TEOF (theophylline), CAF (caffeine), CAT (catechin), and EPICAT (epicatechin). * The initial condition was considered for the samples stored in RE at 24 h of preparation. ** The final condition was considered for the samples stored in RE at 90 days of experiments.

It was not possible to determine the encapsulation/incorporation for TEOB and TEOF because the concentration of these active components in the sample of guarana analyzed was very low: 0.14 μg/mL (TEOB) and 0.47 μg/mL (TEOF). Although the values satisfied the detection limits of the method, they were less than the limits of quantification.

The compounds analyzed in our study exhibited different interactions with the liposomal structures, resulting in different encapsulation/incorporation. There are two kinds of substances that may be stably associated with liposomes: highly water soluble substances and highly lipid soluble substances. In this context, the hydro- or lipophilicity of each active compound will determine whether it will be encapsulated or incorporated into the lipid bilayer. The higher hydrophilicity of CAF compared with that of CAT and EPICAT may justify its lower incorporation into the liposome structure. Similarly, the composition of soy phosphatidylcholine confers higher permeability to the membrane, leading to lower incorporation for compounds with hydrophilic characteristics [70,79].

For CAF, the initial encapsulation/incorporation was 17.02 ± 0.60% but increased to 30.13 ± 0.23% after 90 days of storage under RE. From these results, it is believed that the CAF is free in the dispersion, evincing low encapsulation/incorporation because it is a highly hydrophilic compound. With the passage of time, a greater interaction or permeability may occur with the liposomal system, thereby resulting in better internalization and a subsequent increase in encapsulation/incorporation. This hypothesis was proven when new tests for the encapsulation/incorporation of CAF were performed after 7 days of liposome stability stored under refrigeration. In this period, the encapsulation/incorporation of CAF was 32.61 ± 0.36% and was maintained at 30.13 ± 0.23% until 90 days of stability.

On the other hand, for highly lipophilic materials, such as CAT and EPICAT, when produced by preparation methods using organic solvents, the incorporation into the lipid bilayer is usually close to 100% because these compounds interact with the lipid layers of the liposomes, thus increasing their encapsulation/incorporation [80].

The CAT and EPICAT revealed higher incorporation when compared to the CAF, with values of 74.34 ± 1.93 and 87.53 ± 0.94%, respectively. It should be noted that the incorporation for these compounds was elevated for up to 90 days of stability (CAT, 51.65 ± 0.77%

and EPICAT, 70.88 ± 2.17%) when the samples were stored under RE conditions, and even the total content of these compounds exhibited a significant reduction at 60 days and was ~60% at 90 days under this same condition.

### 3.2. Cytotoxicity Studies

Oxidative stress is one of the main mechanisms contributing to the aging of skin [81]. In this respect, the use of products with potential antioxidant effects can exert beneficial actions on the same, thereby protecting it against aging [82].

In previous studies [37–39,42,44–46,83–85], guarana demonstrated potent antioxidant activity, thereby making it a product of great interest in the cosmetics industry. In studies previously conducted by our research group, it was observed that guarana's antioxidant activity was maintained when it was incorporated into liposomes [49].

Immortalized human keratinocytes are cell lines that retain the capacity for epidermal differentiation. They are the most abundant cells in the epidermis. Therefore, they are kept directly in contact with active substances that are capable of permeating the stratum corneum. Likewise, fibroblast cell lines are the most abundant cell type in the human dermis and allow for the verification of possible damage when the developed product penetrates this layer.

The MTT and NRU assays used for the evaluation of cytotoxicity are based on the color detection of the substances by spectrophotometry and their refraction or light absorption ability. Some nanoparticles may interfere with the spectrophotometric reading system [86–88].

Before the cellular viability experiments, the possible interference of the liposomes with MTT and NR was evaluated. The scanning spectra for the liposome samples were found to be similar to the controls, both for MTT and NR. These results indicated that there is no interference of liposomes with the cell viability techniques used, thus showing reliability in the results obtained. The results for the cell viability of 3T3 cells are presented in Figure 4.

For the 3T3 cells, the NRU assay (Figure 4A,C) demonstrated a decrease in the cell viability at the highest concentration tested (500 μg/mL) for the three different treatments (GL, B-Lip, and G-Lip) after 24 and 48 h of exposure. Moreover, a decrease in cell viability at concentrations of 250 and 500 μg/mL was observed after 72 h. This reduction was visualized for the different treatments, showing no statistically significant differences among them ($p > 0.05$). For GL, after 72 h (Figure 4E), the viability reduction occurred from the concentration of 125 μg/mL (with a final viability of 77.74%). The liposomes (B-Lip and G-Lip) maintained viability higher than 90% at this concentration.

The cell viability determined by the MTT assay (Figure 4B,D,F) exhibited a decrease in the B-Lip cell viability at a concentration of 31.25 μg/mL after 24 h of cellular exposure and remained low after 48 and 72 h. On the other hand, the GL induced a decrease in the cell viability at concentrations of 500 μg/mL (after 24 and 48 h) and 250 μg/mL (after 72 h). The G-Lip viability reduction was only observed at a concentration of 500 μg/mL. The results of the cell viability for HaCaT evaluation are shown in Figure 5.

The cell viability of HaCaT cells determined by the NRU assay demonstrated a decrease of 500 μg/mL after 24 h (GL, 87.36 ± 7.09%) (Figure 5A) and 48 h (GL, 82.63 ± 8.74%; B-Lip, 80.54 ± 7.02%; and G-Lip, 71.02 ± 5.83%) (Figure 5C). When these cells were exposed to the different treatments for 72 h, the viability reduction occurred at a concentration of 250 μg/mL (GL, 88.49 ± 3.62% and B-Lip, 86.38 ± 8.24%) (Figure 5E). On the other hand, at the lowest concentrations assayed (3.91, 7.91, and 15.63 μg/mL), a slight cell proliferation occurred for the liposomes (B-Lip and G-Lip).

The cell viability of the HaCaT cells by the MTT assay showed a decrease of 500 μg/mL after 24 h (B-Lip, 66.41% ± 10.13) (Figure 5B) and 48 h (GL, 75.95 ± 14.91%; B-Lip, 55.41 ± 10.04%; and G-Lip, 65.42 ± 6.30%) for the different treatments (Figure 5D). Similarly, this decrease occurred at a concentration of 125 μg/mL (B-Lip) and 500 μg/mL for the different treatments after 72 h (Figure 5F).

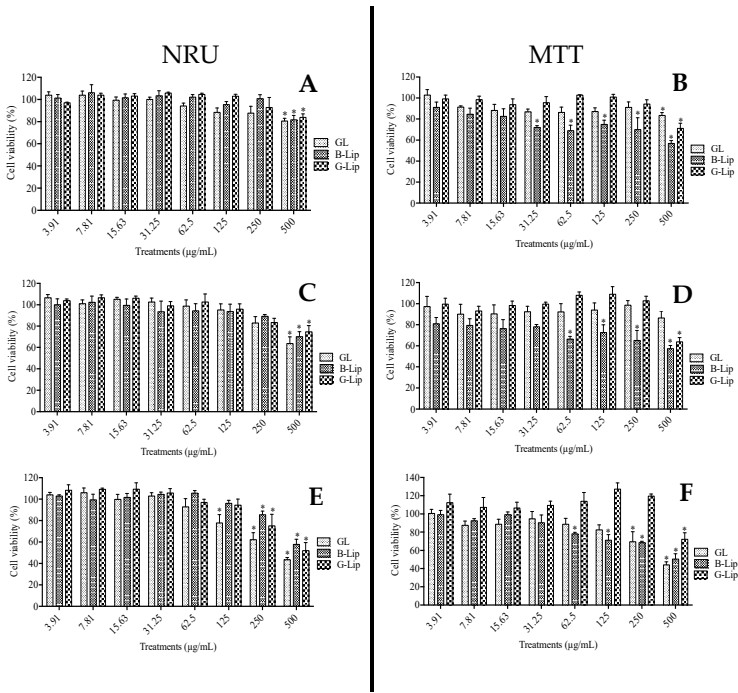

**Figure 4.** In vitro study by cell viability for 3T3 cells. Data were generated from two different protocols: the NRU and MTT techniques at 24 h, 48 h and 72 h. Cell viability by the NRU (**A**,**C**,**E**) and MTT (**B**,**D**,**F**) after 24 h (**A**,**B**), 48 h (**C**,**D**), and 72 h (**E**,**F**), respectively. Three different formulations were analyzed: GL (1 mg/mL guarana powder), B-Lip (blank liposomes), and G-Lip (liposomes containing 1 mg/mL guarana powder). Results are expressed as mean $\pm$ standard deviation (n = 3). * Significant difference ($p < 0.05$) in relation to the control cells (100% viability).

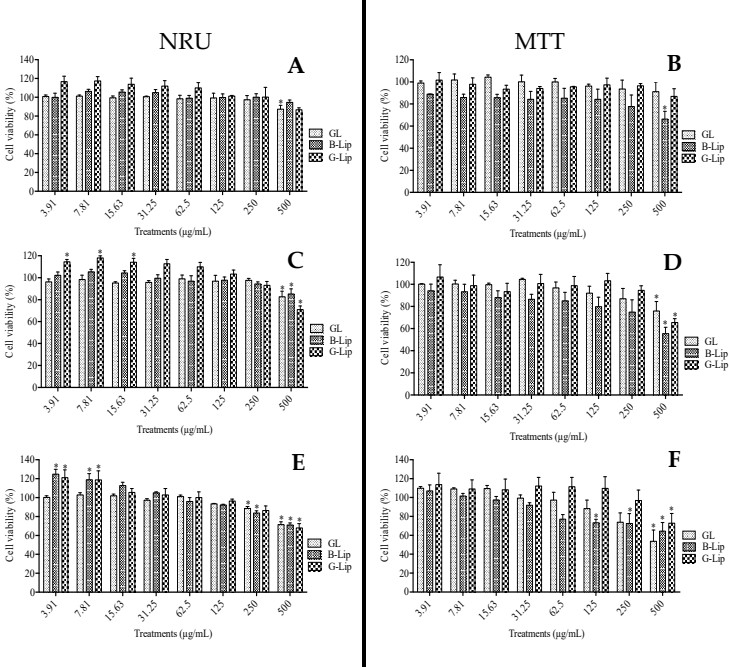

**Figure 5.** In vitro study by cell viability of HaCaT cells. Data were generated from two different protocols: the NRU and MTT techniques at 24 h, 48 h and 72 h. Cell viability measured by the NRU (**A**,**C**,**E**) and MTT (**B**,**D**,**F**) analyzed at 24 h (**A**,**B**), 48 h (**C**,**D**) and 72 h (**E**,**F**), respectively. Three different formulations were analyzed: GL (1 mg/mL guarana powder), B-Lip (blank liposomes), and G-Lip (liposomes containing 1 mg/mL guarana powder). Results are expressed as mean $\pm$ standard deviation (n = 3). * Significant difference ($p < 0.05$) in relation to control cells (100% viability).

The results of the human fibroblast cells treated with guarana (GL) and liposomes (B-Lip and G-Lip) are shown in Figure 6.

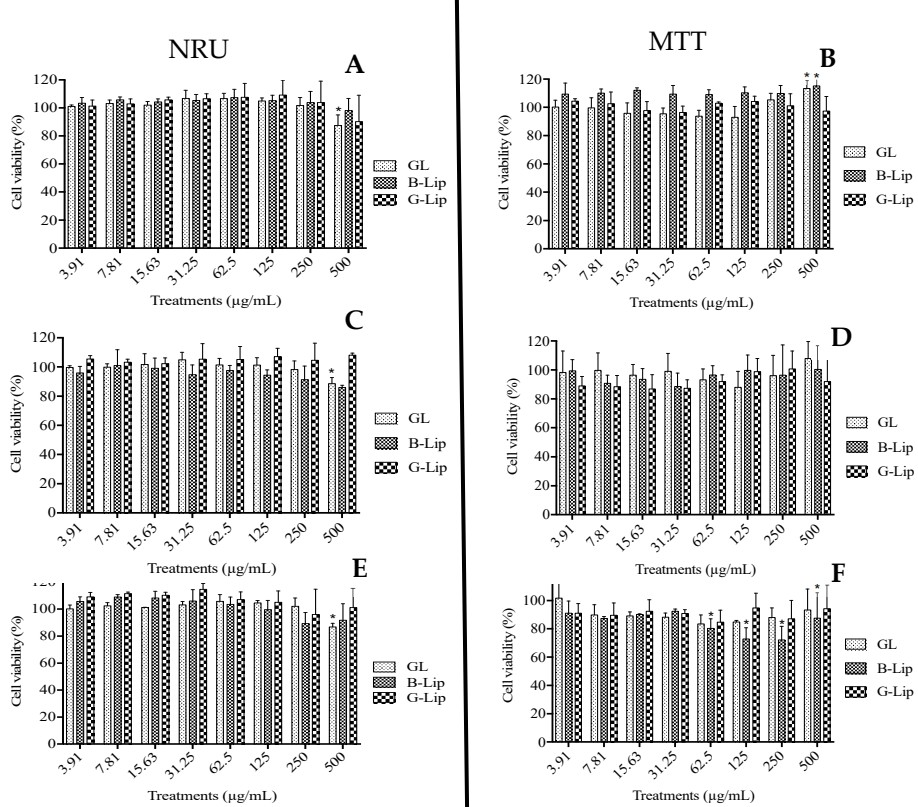

**Figure 6.** In vitro study by cell viability of human fibroblasts cells. Data were generated from two different protocols: the NRU and MTT techniques at 24 h, 48 h and 72 h. Cell viability measured by the NRU (**A**,**C**,**E**) and MTT (**B**,**D**,**F**) analyses at 24 h (**A**,**B**), 48 h (**C**,**D**), and 72 h (**E**,**F**), respectively. Three different formulations were analyzed: GL (1 mg/mL guarana powder), B-Lip (blank liposomes), and G-Lip (liposomes containing 1 mg/mL guarana powder). Results are expressed as mean ± standard deviation (n = 3). * Significant difference ($p < 0.05$) in relation to control cells (100% viability).

Changes in the cell viability determined by the NRU assay were only observed for human fibroblasts at the highest concentration (500 μg/mL) for GL, with a slight reduction (13.16%) after 72 h (Figure 6E).

The MTT assay (Figure 6B,D,F) demonstrated significant changes ($p < 0.05$) after 72 h from a concentration of 62.5 μg/mL for B-Lip (Figure 6F). It was also observed that, after 24 h (Figure 6B), GL and B-Lip exhibited a significant increase ($p < 0.05$) at 500 μg/mL, indicating slight cell proliferation.

A comparison of the different cell viabilities assessed by MTT and NRU revealed a higher cytotoxic response by MTT than by NRU, independent of the cell line tested. This observation has already been described by Nogueira et al. (2011) [89] in a prior study. According to the authors, differences in cytotoxic responses may be related to the mechanisms of toxicity exerted by the compounds, involving an initial effect on the metabolic activity of the cells primarily detected by the MTT technique. However, the plasma membrane and the lysosomal compartments may be affected in one phase, exhibiting less damage when analyzed by the NRU technique.

According to Oliveira (2009) [90], the reduction in the cell viability using nanostructures is acceptable at up to 90%. This finding establishes a standard classification for viability, considering the non-cytotoxic percentage change for viability >90%, slightly cytotoxic from 80% to 89%, moderately cytotoxic from 50% to 79%, and highly cytotoxic at less than 50%. Liposomes (B-Lip) have, on average, moderate cytotoxicity.

The results of the cellular viability observed in our study suggest a protective effect of guarana against the damage of cytotoxicity caused by liposomes for the different cell lines tested since a greater reduction in viability was observed in B-Lip when compared to G-Lip and GL.

The possible protective effect against cytotoxic damage evidenced by guarana may be related to the antioxidant activity exerted by their content on phenolic compounds (present in catechins), which are mainly found in guarana seeds. These compounds actively protect the body against the effects of free radicals, helping to prevent diseases [83]. It is worth noting that one of the components of the G-Lip formulation tested here is vitamin E, which has antioxidant activity; however, based on experiments performed previously [49], vitamin E did not express antioxidant activity in the formulation when evaluated by the DPPH method since no antioxidant activity was observed for the control formulation (B-Lip); on the other hand, antioxidant activity was observed for the G-Lip formulation, proving that guarana is responsible for the antioxidant activity observed in the G-Lip formulations.

In previous studies, Peirano et al. (2011) [54] demonstrated that guarana presented 23% higher cellular esterase activity than formulations without guarana, exerting a vitalizing effect on skin fibroblasts.

Basile et al. (2005) [42] evaluated the antioxidant activity of guarana in 3T3 cells by the malondialdehyde test (MDA), following cell damage by the ferric ammonium citrate (FAC) test. A reduction of 62.5% in the lipid peroxidation was observed when 2 μg/mL guarana concentrations were used, given that it is dose dependent. Likewise, the antioxidant potential was correlated with the presence of phenolic compounds.

Also, in another study, Bittencourt et al. (2013) [38] determined the protective effect of guarana extract by the MTT technique after exposing fibroblast cells (NIH-3T3) to sodium nitroprusside (SNP, 10 μM) for 6 h. The assay was conducted at a concentration that was able to decrease by >90% the cellular viability of 3T3 cells. With the addition of guarana extract at concentrations of 0.5, 1, 5, 10, and 20 mg/mL, the authors observed that guarana was able to reverse the SNS toxicity, especially at lower concentrations (<5 mg/mL), indicating a protective effect of this compound.

## 4. Conclusions

This study describes the stability of liposomes containing 1 mg/mL guarana powder and produced by reverse phase evaporation. These liposomes revealed physicochemical characteristics suitable for the type of nanostructure under study and demonstrated stability for 60 days when the formulations were stored under refrigeration conditions (RE $\pm$ 5 °C). The in vitro cytotoxicity studies for skin cells, 3T3, HaCaT, and human fibroblasts demonstrated a small reduction in cell viability. However, the reduction in cell viability for B-Lip was greater compared to those for GL and G-Lip, thus evidencing possible protection by guarana from cytotoxic effects. In this sense, guarana-loaded liposomes present a potential application for topical administration.

**Author Contributions:** Methodology: I.R., A.J.F.D., P.G., A.F.O., M.M. and M.P.V.; writing—original draft preparation: I.R. and A.J.F.D.; writing—review and editing: I.R., P.G., A.F.O., M.M. and M.P.V.; project administration: I.R., P.G. and M.P.V.; supervision: P.G., A.F.O., M.M. and M.P.V.; funding acquisition: P.G., I.B.M.d.C., E.E.R. and M.P.V. All authors have read and agreed to the published version of the manuscript.

**Funding:** This research was funded by Coordenação de Aperfeiçoamento de Pessoal de Nível Superior—Brazil (CAPES)—Finance Code 001 and CNPq/JSTproject (Process: 490760/2013-9) for financial support. This study was also financed by project 307629 of Fundació Bosch & Gimpera—Universitat de Barcelona.

**Data Availability Statement:** The data that support the findings of this study are available from the corresponding author, [M.P.V], upon reasonable request.

**Acknowledgments:** The authors are thankful to the Funding Sources Coordenação de Aperfeiçoamento de Pessoal de Nível Superior—Brazil (CAPES), CNPq/JSTproject for financial support and Fundació Bosch & Gimpera—Universitat de Barcelona.

**Conflicts of Interest:** The authors declare that they have no known competing financial interests or personal relationships that could have appeared to influence the work reported in this paper.

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
