# Peer review of "Profiling and Evaluation of the Effect of Guarana-Loaded Liposomes on Different Skin Cell Lines: An In Vitro Study"

_cosmetics, doi:10.3390/cosmetics10030079_

Round 1

Reviewer 1 Report

The manuscript submitted by Roggia and co-workers evaluates the physicochemical stability of liposomes containing guarana powder by reverse phase evaporation. Moreover, based on the potential topical applications of these new nanocarriers, the in vitro cytotoxicity of guarana (GL), the blank liposome (without guarana, B-Lip), and the liposome containing 1 mg/mL guarana powder (G-Lip) were tested. The evaluations were conducted in different cultures of skin cells, fibroblasts (3T3 and 1BR.3.G) and keratinocytes (HaCaT) as an alternative to the use of animals-

The main results reveal potential protection by guarana from cytotoxic effects in skin cells and a potential application for topical administration of guarana-loaded liposomes.

The topic is interesting and fits the journal's general scope. The MS is well-written and organised. The methodology section includes all the information needed to make the experiments reproducible by potential readers. The results displayed in the figures are well supported by the text.

Just minor issues related to one of the figures and the format of the references are worth mentioning.

Comments have been embedded through the MS in order to help the authors to improve this version.

Minor spelling and grammar issues must be reviewed.

Author Response

We have upload the answer to the reviewer

Reviewer 2 Report

1. It's so strange that the authors used the format of "Diagnostics".

2. Manuscript title: It should be more direct, maybe L15-16 is more suitable for a manuscript title.

3. M&M: How do the authors know to use 1 mg/mL of guarana, why not test other dosages?

4. Introduction: It's strange that each paragraph is very short in this section and other sections.

5. Avoid using first-person writing throughout the manuscript.

6. Statistical analysis: How many replicates were conducted in each treatment? Also, provide the information in a note for every table and figure.

7. Provide high-quality pictures of the detailed structure and morphology of the liposomes and the cells of the control and the treated ones because Figure 2 is unsatisfactory.

8. Why did the authors choose the three cell lines for this study? Discuss it more in the text.

Author Response

We hava uploaded answer to the reviewer

Round 2

Reviewer 2 Report

All comments have been addressed well, so I have no further questions.